# Impact of Smoking and Obesity on the Selected Peptide Hormones and Metabolic Parameters in the Blood of Women with Polycystic Ovary Syndrome—Preliminary Study

**DOI:** 10.3390/ijms25168713

**Published:** 2024-08-09

**Authors:** Justyna Niepsuj, Agnieszka Piwowar, Grzegorz Franik, Anna Bizoń

**Affiliations:** 1Department of Toxicology, Faculty of Pharmacy, Wroclaw Medical University, 50-556 Wroclaw, Poland; justyna.niepsuj@student.umw.edu.pl (J.N.); agnieszka.piwowar@umw.edu.pl (A.P.); 2Department of Endocrinological Gynecology, Medical University of Silesia, 40-752 Katowice, Poland; gfranik@sum.edu.pl

**Keywords:** polycystic ovary syndrome, peptide hormones, smoking

## Abstract

We investigated the effects of tobacco smoke exposure and abnormal body weight on selected peptide hormones and their association with metabolic and hormonal disorders in women with polycystic ovary syndrome (PCOS). The study group included 88 women with PCOS and 28 women without the disease. In women with PCOS, chemerin, lipocalin, and apelin concentrations were influenced by overweight and obesity status, with the highest concentrations observed in those with a body mass index (BMI) ≥ 30.0. Exposure to tobacco smoke significantly increased only lipocalin-2 concentration. The disease itself did not affect the concentrations of chemerin, lipocalin, and apelin. Additionally, we found a positive correlation between chemerin concentration and fasting glucose, fasting insulin, and triglycerides levels, while a negative correlation was observed with high-density lipoprotein (HDL-C) concentration. In the smoking subgroup, chemerin concentration was positively correlated with free testosterone concentration and the free androgen index and negatively associated with sex hormone-binding globulin concentration. Our findings indicate that abnormal body weight has a stronger impact than tobacco smoke exposure on metabolic and hormonal disorders in women with PCOS, highlighting the important role of weight control in such individuals. However, smoking appears to be an additional factor that intensifies hormonal disorders associated with adipose tissue.

## 1. Introduction

The issue of female infertility in one of the main causes of decreased population growth in developed countries worldwide [1]. One of the leading causes of female infertility is polycystic ovary syndrome (PCOS), which affects 8–13% of women of reproductive age [2,3] and presents with both fertility and metabolic disorders [4]. Epidemiological studies show that approximately 50% of women with PCOS experience obesity and metabolic syndrome [5,6]. Additionally, obesity exacerbates PCOS [7]. Women with PCOS have a higher prevalence of conditions such as insulin resistance (IR), with a prevalence of 35–80%, type 2 diabetes (T2D), cardiovascular diseases, and infertility compared to the general female population [8,9]. These conditions can also be intensified by obesity.

Adipose tissue is a highly active endocrine organ that, through the action of secreted hormone-like compounds, can influence hormonal and metabolic status [10]. Among the molecules predominantly released by adipocytes are chemerin [11], apelin [12], and lipocalin-2 [13], which are involved in many metabolic and hormonal processes. 

Previous studies have shown that chemerin, apelin, and lipocalin-2 regulate insulin secretion, with changes in their levels observed in patients with diabetes [14,15,16]. Additionally, chemerin and lipocalin-2 participate in inflammatory processes [17,18], and the levels of chemerin and apelin may be associated with sex hormone homeostasis. An experimental study conducted on female mice showed that chemerin/G protein-coupled receptor 1 signaling regulates the secretion of prostaglandin-induced progesterone by ovarian cells during follicular development, corpus luteum formation, and luteolysis [19]. Furthermore, increased chemerin concentration can decrease insulin-like growth factor-1-stimulated progesterone and estradiol production in human granulosa cells [20], while apelin regulates the endocrine function of the human placenta [21]. Incorrect expression of lipocalin-2 is related to diseases of the reproductive system [22]. In the female reproductive system, lipocalin-2 plays a pivotal role in tissue reorganization during the menstrual cycle and pregnancy, with higher concentrations observed in the plasma of pregnant women, especially those who are overweight or obese, correlating with markers of IR [23]. As a multifaceted protein, chemerin is expressed in the ovary, placenta, and testis, indicating its potential involvement in normal female reproduction as well as in reproductive system disorders, including PCOS [24]. However, the exact mechanisms of apelin, lipocalin-2, and chemerin, as well as the specific cells that produce them in the course of PCOS, remain unidentified.

Another crucial environment and lifestyle factor affecting human fertility is cigarette smoking. Smoking remains one of the most common legal addictions worldwide and contributes to numerous systemic heath issues, including those affecting the reproductive system [25,26]. The association between smoking and reduced female fecundity and fertility, particularly linking it to primary tubal infertility, has been clearly confirmed [27]. Compounds in tobacco negatively affect ovarian follicle maturation, ovarian reserve, sex steroids synthesis, and embryo development [28]. 

This study builds on previous research conducted on women with PCOS, which demonstrated significant associations between metabolic and hormonal disorders [29], abdominal obesity [30], adipose tissue distribution [31], oxidative stress severity [32], and the harmful effects of tobacco smoke on female fertility [33,34,35,36]. 

The goal of the current study is to investigate the effect of exposure to tobacco smoke and abnormal body weight on selected peptide hormones and their association with metabolic and hormonal parameters in women with PCOS. To the best of our knowledge, the effect of exposure to tobacco smoke on the concentration of chemerin, apelin, and lipocalin-2 in women with PCOS has not yet been evaluated.

## 2. Results

The studied groups had comparable values in terms of age, body mass index (BMI), and waist-to-hip ratio (WHR). Significant differences between the groups were observed in the cases of acne (using the US FDA AGSS scale) and hirsutism (a modified Ferriman–Gallwey method (mF-G)) intensity. Increased concentrations of fasting glucose and insulin, as well as increased values of homeostatic model assessment for insulin resistance (HOMA-IR), were found in the group of women with PCOS compared to the control group. Furthermore, in the group of women with PCOS, increased concentrations of dehydroepiandrosterone sulfate (DHEA-S), total testosterone (tTest), and free testosterone (fTest) and increased values of the free androgen index (FAI) were detected, while the concentrations of high-density lipoprotein (HDL-C) and sex hormone-binding globulin (SHBG) were decreased. We did not reveal any significant differences in apelin, chemerin, and lipocalin-2 concentrations between women with PCOS and the control group (Table 1). 

The evaluation of exposure to tobacco smoke revealed increased concentrations of lipocalin-2 and triglycerides and increased value of FAI, whereas there were decreased concentrations of HDL-C and SHBG in the blood of women with PCOS compared to the control group. No significant changes were found in the case of other parameters (Table 2).

We divided the group of women with PCOS according to normal weight (BMI < 25.0), overweight (BMI ≥ 25.0), and obesity (BMI ≥ 30.0). Significant changes were observed in all studied peptide hormones—apelin, chemerin, and lipocalin-2. Increased BMI was associated with decreased concentrations of HDL-C and SHBG and increased concentrations of triglycerides, fasting glucose, fasting insulin, tTest, and fTest, as well as increased HOMA-IR and FAI values. Significant alterations were also found in the concentration of anti-Müllerian hormone (AMH) between those subgroups (Table 3). 

Interestingly, visceral obesity was not associated with significant changes in the concentrations of apelin, chemerin, or lipocalin-2. Increased concentrations of low-density lipoprotein (LDL-C), triglycerides, fasting insulin, DHEA-S, tTest, and fTest and increased values of HOMA-IR and FAI were observed in women with PCOS and WHR ≥ 0.8 compared to women with PCOS and WHR < 0.8. A higher WHR was related to decreased concentrations of HDL-C and SHBG (Table 4).

Considering the significant correlations in the entire group of women with PCOS among the concentrations of apelin, chemerin, and lipocalin-2 and the studied parameters, we found the most significant relationships with the concentration of lipocalin-2, while no significant association was observed with apelin concentration. Chemerin concentration was positively correlated with BMI and WHR values, as well as with the concentrations of triglycerides, fasting glucose, and insulin, but negatively correlated with HDL-C concentration (Table 5). 

We also analyzed the correlation coefficients in the group of smoking women with PCOS. In the smoking women with PCOS, all significant correlation coefficients had higher values than in the entire group of women. Unexpectedly, the most significant correlations were found with the concentration of chemerin. Apelin concentration was negatively associated with the concentration of fasting insulin and the value of HOMA-IR while positively associated with the concentration of SHGB. Lipocalin-2 concentration was negatively associated with the values of the mF-G scale and follicle-stimulating hormone (FSH), as well as SHBG concentration, while positively associated with AD concentration.

Chemerin concentration was positively associated with the concentrations of total cholesterol, LDL-C, triglycerides, fasting glucose, fasting insulin, and tTest, as well as with the values of WHR, HOMA-IR, and FAI, while being negatively associated with the concentration of HDL-C and SHBG (Table 6).

## 3. Discussion

The associations between fat tissue and metabolic or hormonal homeostasis disorders in the course of PCOS are clearly confirmed [37]. Furthermore, exposure to tobacco smoke could additionally intensify these disorders. In the present study, we aimed to investigate the impact of the magnitude of hormonal and metabolic disorders in the blood of non-smoking and smoking women with PCOS who are overweight or obese. We estimated the relationships between BMI or WHR values and hormonal disorders, particularly with the concentration of chemerin, lipocalin-2, and apelin, peptide hormones mainly produced by adipocytes. 

Although significant changes in the concentrations of chemerin, lipocalin, and apelin were not observed between women with and without PCOS, those parameters were altered by overweight or obesity status, as indicated by BMI, in the group of women with PCOS. However, no significant changes were found when the women with PCOS were divided based on WHR values of <0.80 or ≥0.80, which might suggest that general overweight/obesity status but not the type of obesity affects these concentrations. 

On the other hand, when we evaluated the relationships between WHR and chemerin, lipocalin, and apelin, the concentration of chemerin was significantly associated with WHR in both the entire group of women with PCOS and the smoking subgroup of women with PCOS. BMI was positively correlated with chemerin concentration in the entire group of women with PCOS and negatively correlated with apelin concentration in the smoking subgroups of women with PCOS. Increased chemerin concentration in obese women was revealed in a study conducted by Guzel et al. [38], which compared obese women to lean women. These findings are consistent with the present study, as lower chemerin concentrations were detected in the serum of women with normal weight, while women with obesity had the highest concentrations. 

A meta-analysis conducted by Wang et al. [39] showed higher chemerin levels in the blood of women with PCOS than in healthy women. The authors also observed a positive relationship among chemerin concentration, obesity, infertility prevalence, HOMA-IR value, and total testosterone or glucose levels, suggesting that chemerin could be involved in the pathogenesis of PCOS. 

Analyzing the association between determined peptide hormones and studied parameters in the subgroup of smoking women with PCOS, the most significant correlations were detected between chemerin concentration and lipid or glucose parameters and selected hormones. Generally, we found a positive correlation between chemerin concentration and fasting glucose, fasting insulin, and triglycerides level, while a negative correlation was observed with HDL-C concentration. 

These observations were revealed both in the entire group and, more strongly, in the smoking subgroup of women with PCOS. Furthermore, in the smoking subgroup, the concentration of chemerin was positively correlated with fTest concentration and FAI value while being negatively associated with SHBG concentration. The relationship of chemerin with hyperandrogenemia was also presented in experimental studies conducted by Lima et al. [40] and Wang et. al. [41]. Lima et al. [40] noted that increased serum chemerin levels were evident in overweight rats, suggesting that the concentration of chemerin detected in serum may be reflective of a systemic response and associated with obesity. Wang et al. [41], in the study conducted on a hyperandrogenic rodent model, showed that recombinant chemerin suppressed FSH-induced expression of steroidogenic enzymes, including aromatase, a crucial enzyme catalyzing the conversion of androgens into estrogens. Therefore, changes in chemerin concentration could be associated with hyperandrogenism, which is one of the main features of PCOS.

Summarizing the involvement of chemerin in the course of PCOS, we can suppose that the concentration of chemerin is affected by disorders in lipid and glucose profiles and the presence of hyperandrogenism. Exposure to tobacco smoke could be recognized as an additional factor influencing these relationships but not as effectively as the factors mentioned above.

When we considered the concentration of apelin, we did not observe any significant correlations between its concentration and any hormonal or metabolic parameters in the entire group of women with PCOS. However, in the group of smoking women with PCOS, we found a negative correlation between apelin concentration and BMI (r = −0.45), insulin concentration (r = −0.38), and HOMA-IR (r = −0.39). We also checked the concentration of apelin in the group of smoking women with PCOS divided according to the HOMA-IR values of <2.0 (n = 19) or ≥2.0 (n = 11) and revealed a significantly decreased concentration in the group of smoking women with HOMA-IR ≥ 2.0 compared to the group with HOMA-IR < 2.0 (data not included in the tables). Even though we detected significant differences in apelin concentration, we did not reveal any significant correlations between apelin concentration and sex hormone levels. In contrast, the clinical study performed by Choi et al. [42] on 115 non-obese (BMI < 25 kg/m^2^) Korean women with PCOS found that serum apelin concentration was negatively correlated with tTest levels and the FAI, and these relationships were found to be independent of IR. The differences between our studies might be associated with the characteristics of the studied groups—in Choi’s study, only women with normal body weight were investigated, while in our study, approximately 47% of the women had abnormal weight (BMI ≥ 25.0 or BMI ≥ 30.0). Additionally, our study also included smoking women. Finally, other crucial factors could be related to ethnic, environmental, and geographic differences, which could also affect the investigated parameters. Therefore, further investigations performed on larger groups are needed to better explain the role of apelin in the course of PCOS.

Information about lipocalin-2 in the context of PCOS is limited, and existing research is inconclusive. A study by Cakal et al. [43] reported elevated levels of lipocalin-2 in the serum of women with PCOS compared to a control group. Conversely, research conducted by Panidis et al. [44], Koiou et al. [45], and Halawa et al. [46] did not find any significant changes between women with and without PCOS. However, these studies [44,45,46] noted that weight loss significantly reduced lipcalin-2 levels in overweight or obese women with PCOS, including those taking metformin [46]. Similar to chemerin, lipocalin-2 levels were dependent on weight. The lowest concentration was found in women with BMI < 25.0, while the highest was in women with BMI ≥ 25.0.

It also appears that only the lipocalin-2 concentration was altered by exposure to tobacco smoke, which was manifested by a higher concentration in the subgroup of smoking women with PCOS compared to non-smoking women with PCOS. Additionally, there was a low but significant positive correlation (0.29; *p* < 0.008) between the concentration of cotinine and lipocalin-2, which further confirmed this association. 

We would also like to highlight the association between lipocalin-2 and FSH concentration detected in the present study. FSH plays a pivotal role in fertility and is used clinically to treat women with anovulatory infertility [47]. Among studied peptide hormones, only lipocalin-2 was significantly correlated with FSH concentration, and these relationships were detected both in the entire group of women with PCOS as well as in the subgroups of smoking women, with higher values of coefficient correlation noted in the smoking subgroup. Furthermore, only the concentration of lipocalin-2 was positively correlated with the presence of hirsutism, which may also confirm the involvement of lipocalin-2 in hormonal disorders present in the course of PCOS. Summarizing this part of the present study, the most significant correlations in the entire group of women with PCOS were detected between lipocalin-2 and the studied parameters, which puts this peptide hormone in the limelight of our research.

In the present study, we found that all investigated factors, such as smoking, higher BMI, and WHR values, decreased SHBG concentration. The concentration of SHBG, a pivotal globulin that binds estrogen and androgens and regulates their biological activities, is used as an indicator of hyperandrogenism in women with PCOS [48]. Additionally, decreased SHBG concentration during PCOS is considered a marker of abnormal glucose and lipid metabolism and is associated with the long-term prognosis of PCOS [48]. We also noted a significant correlation between SHBG concentration and apelin, chemerin, and lipocalin-2 concentrations in the smoking women, with the strongest value of coefficient correlation between SHBG and lipocalin-2 concentration. This correlation was also observed in the entire group of women with PCOS. In all cases, the correlation coefficients were negative. Multiple regression analysis performed on a cohort of 606 women with PCOS in our previous published study [29] revealed that SHBG concentration and FAI value were the main parameters associated with metabolic and biochemical disruptions in the course of PCOS. Taken together, significantly decreased levels of SHBG may indicate elevated levels of free androgens, which was observed in the present study as increased fTest concentrations and FAI values in women with PCOS compared to women without PCOS. Simultaneously, increased levels of apelin, chemerin, and lipocalin-2 suggest complex metabolic disorders that may be associated with obesity and IR. Different or stronger values of coefficient correlations observed in the smoking subgroup of women with PCOS than in the entire group of women with PCOS suggest that exposure to tobacco smoke induces further disturbances in the course of PCOS. 

Our findings indicate that abnormal body weight has a stronger impact than tobacco smoke exposure on metabolic and hormonal disorders in women with PCOS, highlighting the important role of weight control in such individuals. However, smoking appears to be an additional factor that intensifies hormonal disorders associated with adipose tissue. 

Furthermore, our results clearly confirm the complexity and heterogeneous nature of PCOS and the difficulty in unambiguously estimating the impact of individual etiological factors in the course of this disease. 

We also highlight that due to the relatively small number of patients, our results should be treated as a preliminary. 

Figure 1 summarizes the main results observed in the entire group of women with PCOS. 

## 4. Materials and Methods

The study group consisted of 116 women, of whom 88 were diagnosed with PCOS, while 28 women were identified as not having this disease. The diagnosis of PCOS was established based on the Rotterdam criteria [49], requiring at least two of the following three conditions: ovulatory dysfunction (OD) (menstrual cycles longer than 35 days or amenorrhea), clinical or biochemical signs of hyperandrogenism (HA), and polycystic ovarian morphology (PCOM) on ultrasound. The diagnosis was made after ruling out other causes of hyperandrogenism, such as congenital adrenal hyperplasia, Cushing’s syndrome, or virilization. Additionally, four different phenotypes of PCOS were recognized according to the Rotterdam criteria: phenotype 1—with OD, HA, PCOM; phenotype 2—with OD, HA; phenotype 3—with HA, PCOM; and phenotype 4—with OD, PCOM. The frequencies of these phenotypes were 65% for phenotype 1; 10% for phenotype 2; 19% for phenotype 3; and 6% for phenotype 4.

Within the PCOS patient group, women were further divided based on their exposure to tobacco smoke: 57 were non-smokers, and 31 were smokers. Biological material for the research was obtained through collaboration with the Department of Gynecological Endocrinology at the Silesian Medical University in Katowice, as part of the research project titled “Molecular Studies Useful in the Assessment of Metabolic Disorders in Patients with Polycystic Ovary Syndrome,” Registration Number CWN UMW: BW-52/2020, established on 17 May 2020. Ethical approval for the research was obtained from the Bioethical Committee of Wroclaw Medical University, Poland (KBN No. 254/2021), and was conducted in accordance with the Helsinki Declaration.

The blood samples were collected during the follicular phase (within 3 and 5 days of the menstrual cycle).

### 4.1. Diagnostic Parameters

During hospitalization, measurements were taken to describe metabolic and hormonal parameters, including total cholesterol, LDL-C, HDL-C, triglycerides, fasting insulin and glucose, SHBG, FSH, luteinizing hormone (LH), fTest, tTest, androstenedione (AD), AMH, and DHEAS concentrations.

Lipid and glucose levels were determined using a colorimetric method from Roche (Basel, Switzerland).

The concentrations of insulin, SHBG, FSH, LH, tTest, tTest, AD, and DHEA-S were measured using ELISA methods provided by DRG Instruments GmbH (Marburg, Germany). 

The levels of AMH were assayed using a commercial ELISA kit conducted by Immunotech a.s. (Prague, Czech Republic). 

Additionally, FAI and HOMA-IR values were calculated according to the following formulas, respectively:FAI=tTestnmolL∗100SHBGnmolL
HOMA−IR=fasting glucosemmolL∗fasting insulinmUmL22.5

### 4.2. Anthropometric Parameters, Acne, and Hirsutism Scale

Standardized methods were used for anthropometric measurements, including height, weight, waist circumference, and hip circumference at the hospital. BMI and WHR were calculated using standard formulas.

Acne was evaluated using the US FDA AGSS scale for a five-category global system [50]. The five categories were graded according to the following values:Clear—no inflammatory or non-inflammatory lesions.Almost clear—rare non-inflammatory lesions (≤one papule).Mild—some non-inflammatory lesions (≤few papules, no nodules).Moderate—many non-inflammatory lesions, some inflammatory lesions, no more than one small nodule.Severe—up to many non-inflammatory and inflammatory lesions, a few nodules.

Using a modified Ferriman–Gallwey method (mF-G), we assessed the intensity of hirsutism, which was defined as a sum score of ≥8. The total range of the score is from 0 to 36 [51,52].

### 4.3. Other Parameters

According to the manufacturer’s instructions, the following parameters were measured using the enzyme-linked immunoassay (ELISA) method, including cotinine (Cotinine ELISA, Cat. No: CO096D, CALBIOTECH, El Cajon, CA, USA); chemerin (Chemerin ELISA kit, Cat. No E102, Mediagnost, Reutlingen, Germany); apelin (Human Apelin 36 ELISA Kit Cat. No GR111150-1, Genorise Scientific Inc., Glen Mills, PA, USA); and lipocalin-2 (Human Lipocalin-2 Assay Kit, Cat. No EL3510-1, AssayPro, Saint Charles, MO, USA).

### 4.4. Statistical Analysis

The concentrations or values of the studied parameters were expressed as means ± SD and medians (first quartile, third quartile). The Shapiro–Wilk test was used to check the normality of the variables, and Levene’s test was applied to assess the homogeneity of variance. When the assumptions of normal distribution and equal variance were not met, differences between two groups (women without vs. with PCOS, non-smoking vs. smoking, women with WHR < 0.8 vs. ≥0.8) were evaluated using the non-parametric Mann–Whitney test. For differences among three groups (divided according to BMI), the Kruskal–Wallis one-way analysis of variance by ranks was used. Correlations were determined using Spearman’s rank-order correlation coefficient. A significant level of *p* < 0.05 was applied to all analyses. Statistical calculations were carried out using Statistica Software Package, version 13.3 (StatSoft, Kraków, Poland).

## Figures and Tables

**Figure 1 ijms-25-08713-f001:**
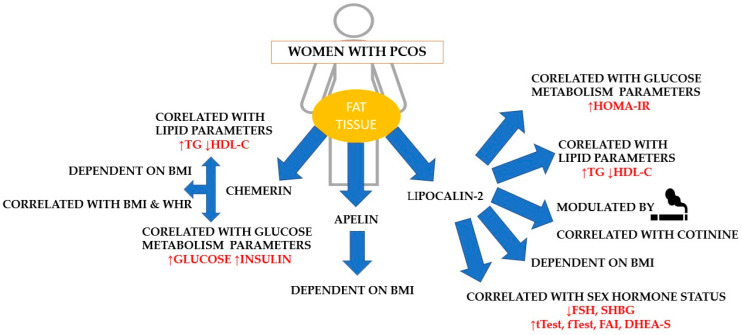
The main relationships among chemerin, apelin, or lipocalin-2 concentration and anthropometric, metabolic, or sex hormone parameters or smoking. Legend: PCOS—polycystic ovary syndrome; BMI—body mass index; HDL-C—high-density lipoprotein; TG—triglycerides; HOMA-IR—homeostatic model assessment for insulin resistance; FSH—follicle-stimulating hormone; DHEA—S-dehydroepiandrosterone sulfate; SHBG—sex hormone-binding globulin; tTest—total testosterone; fTest—free testosterone; FAI—free androgen index.

**Table 1 ijms-25-08713-t001:** Comparison of the studied parameters between women with PCOS and control group.

Parameters	Women with PCOS	Control Group	*p* Value
	n = 88	n = 28	
Age (years)	26.40 ± 4.52 24.00 (21.00–27.00)	25.72 ± 6.10 25.00 (21.00–31.00)	NS
BMI (kg/m^2^)	25.63 ± 6.39 23.92 (20.75–30.46)	26.89 ± 24.03 21.80 (18.34–26.70)	NS
WHR	0.79 ± 0.09 0.77 (0.72–0.84)	0.76 ± 0.08 0.73 (0.70–0.80)	NS
Cotinine (ng/mL)	16.46 ± 24.08 0.41 (0.01–34.68)	5.74 ± 14.35 0.41 (0.02–0.88)	NS
mF-G (0–36)	5.59 ± 4.63 5.00 (1.00–9.00)	4.22 ± 5.54 2.00 (0.00–7.00)	0.033
Acne (1–5)	3.60 ± 12.47 2.00 (1.00–2.00)	0.88 ± 0.87 1.00 (0.00–1.00)	0.000
Apelin (pg/mL)	96.08 ± 454.00 32.57 (30.11–37.91)	80.37 ± 202.78 33.84 (27.76–39.19)	NS
Chemerin (ng/L)	128.7 ± 57.80 120.52 (79.51–156.80)	117.48 ± 55.40 118.34 (78.31–150.83)	NS
Lipocalin-2 (ng/mL)	217.35 ± 84.66 207.07 (153.42–276.68)	189.41 ± 71.44 180.58 (151.77–213.51)	NS
Cholesterol (mg/dL)	163.23 ± 30.36 159.50 (141.00–187.00)	167.56 ± 32.87 163.00 (144.00–192.00)	NS
HDL-C (mg/dL)	55.50 ± 12.34 54.65 (48.05–63.25)	65.16 ± 14.25 63.85 (57.10–72.25)	0.000
LDL-C (mg/dL)	87.00 ± 25.23 83.42 (67.06–104.92)	87.31 ± 28.42 88.42 (73.51–103.47)	NS
Triglycerides (mg/dL)	103.78 ± 57.60 83.60 (63.15–130.00)	79.68 ± 30.77 70.80 (54.80–95.95)	NS
Glucose 0’ (mg/dL)	85.04 ± 6.20 84.50 (80.50–89.05)	81.64 ± 5.31 80.95 (78.40–86.20)	0.015
Insulin 0’ (mU/mL)	9.03 ± 6.44 7.03 (5.16–11.30)	5.27 ± 2.43 4.93 (3.55–7.10)	0.000
HOMA-IR	1.99 ± 1.56 1.50 (1.08–2.40)	1.07 ± 0.52 0.98 (0.68–1.44)	0.000
LH (lU/L)	8.74 ± 6.44 7.18 (5.58–9.90)	9.49 ± 17.66 5.42 (5.51–8.98)	NS
FSH (lU/L)	6.16 ± 1.23 6.03 (5.45–6.93)	6.09 ± 2.32 6.25 (5.21–7.33)	NS
DHEA-S (µg/mL)	321.03 ± 142.64 290.00 (222.00–372.00)	257.26 ± 132.31 227.00 (172.00–314.50)	0.029
SHBG (nmol/L)	53.96 ± 33.25 47.35 (29.75–70.90)	65.15 ± 32.88 59.98 (45.75–71.85)	0.035
tTest (ng/mL)	1.40 ± 0.633 1.23 (0.95–1.78)	1.26 ± 1.26 0.98 (0.65–1.26)	0.011
fTest (pg/mL)	3.02 ± 2.66 2.02 (1.45–3.82)	2.43 ± 2.56 1.45 (0.76–2.81)	0.047
AD (ng/mL)	2.48 ± 0.98 2.19 (1.80–2.91)	2.59 ± 2.70 1.87 (1.56–2.73)	NS
FAI	3.91 ± 3.62 2.66 (1.65–4.40)	2.33 ± 2.43 1.48 (0.88–2.43)	0.003
AMH (ng/mL)	6.34 ± 3.50 5.92 (3.63–8.37)	3.62 ± 2.80 2.95 (1.43–5.00)	0.000

Legend: PCOS—polycystic ovary syndrome; BMI—body mass index; mF-G—Ferriman–Gallwey scale; HDL-C—high-density lipoprotein; LDL-C—low-density lipoprotein; HOMA-IR—homeostatic model assessment for insulin resistance; LH—luteinizing hormone; FSH—follicle-stimulating hormone; DHEA—S-dehydroepiandrosterone sulfate; SHBG—sex hormone-binding globulin; tTest—total testosterone; fTest—free testosterone; AD—androstenedione; FAI—free androgen index; AMH—anti-Müllerian hormone, NS—not significant.

**Table 2 ijms-25-08713-t002:** The influence of tobacco smoke on studied parameters.

Parameters	Women with PCOS		*p* Value
	Non-Smoking n = 57	Smoking n = 31	
Age (years)	24.15 ± 4.35 24.00 (21.00–27.00)	24.83 ± 4.79 24.00 (21.00–28.00)	NS
BMI (kg/m^2^)	25.17 ± 6.12 23.66 (20.42–29.49)	26.71 ± 6.86 25.28 (21.01–30.49)	NS
WHR	0.79 ± 0.10 0.78 (0.71–0.85)	0.78 ± 0.07 0.76 (0.73–0.82)	NS
Cotinine (ng/mL)	0.47 ± 0.24 0.38 (0.09–0.42)	45.23 ± 18.08 49.45 (30.75–60.42)	0.000
mF-G (0–36)	5.65 ± 4.93 4.00 (1.00–9.00)	5.26 ± 4.04 5.00 (2.00–8.00)	NS
Acne (1–5)	4.74 ± 15.37 2.00 (1.00–2.00)	1.45 ± 1.03 2.00 (1.00–2.00)	NS
Apelin (pg/mL)	33.59 ± 6.25 32.57 (30.11–36.46)	36.35 ± 10.99 31.33 (30.11–40.59)	NS
Chemerin (ng/L)	126.45 ± 60.22 119.74 (74.11–154.77)	131.93 ± 53.05 126.01 (86.25–157.29)	NS
Lipocalin-2 (ng/mL)	200.96 ± 82.40 199.08 (142.15–253.04)	248.00 ± 80.00 237.73 (178.22–328.14)	0.016
Cholesterol (mg/dL)	166.02 ± 31.26 161.00 (139.00–195.00)	159.20 ± 28.70 157.50 (142.00–180.00)	NS
HDL-C (mg/dL)	57.82 ± 11.86 57.40 (50.90–63.70)	51.85 ± 12.71 49.70 (43.40–62.30)	0.029
LDL-C (mg/dL)	89.04 ± 25.26 88.08 (67.20–105.20	83.67 ± 24.92 80.40 (66.92–103.00)	NS
Triglycerides (mg/dL)	95.77 ± 52.93 81.40 (61.70–111.00)	118.70 ± 62.69 91.10 (66.80–154.00)	0.042
Glucose 0’ (mg/dL)	84.15 ± 5.21 83.80 (79.80–88.90)	86.87 ± 7.49 86.65 (80.90–89.60)	NS
Insulin 0’ (mU/mL)	8.51 ± 5.72 6.83 (4.70–10.00)	9.91 ± 7.52 7.89 (5.47–12.00)	NS
HOMA-IR	1.82 ± 1.27 1.48 (0.98–2.07)	2.20 ± 1.20 1.60 (1.19–2.41)	NS
LH (lU/L)	8.68 ± 7.25 7.15 (5.53–9.51)	8.78 ± 4.56 7.26 (5.62–10.40)	NS
FSH (lU/L)	6.16 ± 1.27 5.94 (5.45–7.11)	6.16 ± 1.18 6.11 (5.35–6.67)	NS
DHEA-S (µg/mL)	300.91 ± 120.80 285.00 (223.00–364.00)	339.40 ± 174.94 293.00 (209.00–446.00)	NS
SHBG (nmol/L)	58.10 ± 33.41 54.70 (32.90–73.00)	46.12 ± 31.49 34.45 (26.10–55.50)	0.023
tTest (ng/mL)	0.39 ± 0.165 0.34 (0.27–0.49)	0.43 ± 0.21 0.37 (0.29–0.57)	NS
fTest (pg/mL)	2.87 ± 2.44 2.02 (1.41–3.67)	3.29 ± 3.04 2.14 (1.50–4.11)	NS
AD (ng/mL)	2.51 ± 0.98 2.18 (1.77–3.16)	2.40 ± 1.01 2.19 (1.79–2.78)	NS
FAI	3.28 ± 2.95 2.32 (1.65–3.77)	4.98 ± 4.39 3.79 (1.83–6.45)	0.040
AMH (ng/mL)	6.66 ± 3.57 6.18 (3.76–8.99)	5.73 ± 3.27 5.50 (3.43–7.65)	NS

Legend: PCOS—polycystic ovary syndrome; BMI—body mass index; mF-G—Ferriman–Gallwey scale; HDL-C—high-density lipoprotein; LDL-C—low-density lipoprotein; HOMA-IR—homeostatic model assessment for insulin resistance; LH—luteinizing hormone; FSH—follicle-stimulating hormone; DHEA—S-dehydroepiandrosterone sulfate; SHBG—sex hormone-binding globulin; tTest—total testosterone; fTest—free testosterone; AD—androstenedione; FAI—free androgen index; AMH—anti-Müllerian hormone, NS—not significant.

**Table 3 ijms-25-08713-t003:** The effect of BMI value in the group of women with PCOS on studied parameters.

Parameters	Women with PCOS			
	BMI < 25.0 n = 47	BMI ≥ 25.0 n = 18	BMI ≥ 30.0 n = 23	*p* Value
Age (years)	24.13 ± 4.80 23.00 (21.00–27.00) 26.00 (23.00–29.00)	23.61 ± 4.72 23.00 (20.00–25.00)	25.61 ± 3.46 26.00 (23.00–29.00)	NS
BMI (kg/m^2^)	20.90 ± 2.07 20.82 (19.96–22.04)	26.52 ± 1.90 25.86 (25.15–28.55)	34.59 ± 3.70 34.23 (31.25–37.18)	0.000
WHR	0.75 ± 0.09 0.74 (0.71–0.78)	0.79 ± 0.07 0.79 (0.76–0.83)	0.85 ± 0.08 0.86 (0.82–0.91)	0.000
Cotinine (ng/mL)	14.07 ± 23.13 041 (0.01–19.41)	18.35 ± 25.13 0.41 (0.41–42.39)	20.26 ± 26.22 0.41 (0.00–37.86)	NS
mF-G (0–36)	5.63 ± 4.81 5.00 (1.00–9.00)	5.78 ± 4.41 6.50 (1.00–9.00)	5.77 ± 4.50 4.50 (2.00–9.00)	NS
Acne (1–5)	5.28 ± 16.60 2.00 (1.00–2.00)	1.11 ± 1.08 1.00 (0.00–1.08)	1.77 ± 1.38 2.00 (1.00–2.00)	NS
Apelin (pg/mL)	37.22 ± 9.07 35.13 (30.11–43.49)	30.75 ± 4.84 30.11 (27.76–32.58)	46.50 ± 65.99 31.33 (27.76–33.84)	0.016
Chemerin (ng/L)	111.14 ± 40.93 106.60 (77.16–136.45)	149.40 ± 69.85 132.54 (101.63–194.62)	151.83 ± 66.98 154.76 (104.94–203.75)	0.037
Lipocalin-2 (ng/mL)	199.34 ± 88.52 190.48 (141.19–244.58)	235.72 ± 75.00 222.47 (186.15–272.71)	245.72 ± 82.62 262.12 (199.37–299.40)	0.023
Cholesterol (mg/dL)	158.78 ± 28.53 154.50 (137.00–180.00)	159.44 ± 25.69 163.00 (144.00–178.00)	176.78 ± 33.69 180.00 (143.00–201.00)	NS
HDL-C (mg/dL)	60.81 ± 11.13 60.05 (52.90–67.20)	53.64 ± 8.50 54.05 (47.80–62.00)	46.40 ± 11.05 45.90 (38.90–51.50)	0.000
LDL-C (mg/dL)	82.66 ± 23.58 81.57 (66.34–101.58)	85.48 ± 21.71 86.46 (67.20–97.50)	98.16 ± 27.86 99.90 (72.50–121.50)	NS
Triglycerides (mg/dL)	76.79 ± 27.89 66.45 (55.80–91.50)	101.61 ± 38.89 89.80 (76.50–110.00)	161.09 ± 70.36 153.00 (92.60–211.00)	0.000
Glucose 0’ (mg/dL)	83.52 ± 4.91 83.60 (79.60–86.70)	84.61 ± 4.86 85.00 (80.90–87.50)	88.48 ± 7.86 87.90 (82.20–92.00)	0.025
Insulin 0’ (mU/mL)	5.90 ± 2.38 5.47 (4.18–7.23)	8.36 ± 2.53 7.67 (6.14–10.00)	15.91 ± 8.37 13.50 (11.20–18.70)	0.000
HOMA-IR	1.22 ± 0.53 1.16 (0.81–0.53)	1.75 ± 0.56 1.61 (1.32–2.07)	3.67 ± 2.09 3.08 (2.41–3.95)	0.000
LH (lU/L)	9.65 ± 7.73 7.73 (6.02–7.73)	6.64 ± 2.50 6.17 (5.38–7.63)	8.62 ± 5.04 7.27 (5.43–10.50)	NS
FSH (lU/L)	6.43 ± 1.15 6.29 (5.71–7.53)	5.91 ± 1.67 5.68 (4.56–7.53)	5.92 ± 0.90 5.68 (5.35–6.23)	NS
DHEA-S (µg/mL)	295.57 ± 127.51 274.50 (207.00–336.00)	355.28 ± 116.63 303.00 (289.00–379.00)	343.09 ± 173.88 298.00 (201.00–473.00)	NS
SHBG (nmol/L)	68.84 ± 35.93 66.50 (45.30–80.70)	43.07 ± 17.56 38.85 (29.60–56.00)	32.07 ± 16.53 26.80 (20.90–44.40)	0.000
tTest (ng/mL)	0.36 ± 0.18 0.34 (0.25–0.43)	0.43 ± 0.17 0.38 (0.29–0.56)	0.47 ± 0.21 0.45 (0.31–0.64)	0.029
fTest (pg/mL)	2.23 ± 1.60 1.71 (1.23–2.61)	2.57 ± 1.65 2.11 (1.30–3.49)	5.07 ± 3.82 4.09 (2.02–6.77)	0.003
AD (ng/mL)	2.52 ± 0.91 2.29 (1.85–3.03)	2.31 ± 0.77 2.07 (1.79–2.57)	2.56 ± 1.25 2.08 (1.65–3.07)	NS
FAI	2.35 ± 1.94 1.96 (1.14–2.82)	4.07 ± 2.96 3.36 (2.11–3.95)	6.89 ± 4.61 5.49 (3.73–9.80)	0.000
AMH (ng/mL)	6.55 ± 3.32 6.21 (3.87–9.34)	4.97 ± 2.59 3.82 (3.19–7.39)	6.60 ± 4.28 5.72 (4.33–7.57)	0.000

Legend: PCOS—polycystic ovary syndrome; BMI—body mass index; mF-G—Ferriman–Gallwey scale; HDL-C—high-density lipoprotein; LDL-C—low-density lipoprotein; HOMA-IR—homeostatic model assessment for insulin resistance; LH—luteinizing hormone; FSH—follicle-stimulating hormone; DHEA—S-dehydroepiandrosterone sulfate; SHBG—sex hormone-binding globulin; tTest—total testosterone; fTest—free testosterone; AD—androstenedione; FAI—free androgen index; AMH—anti-Müllerian hormone, NS—not significant.

**Table 4 ijms-25-08713-t004:** The effect of WHR value in the group of women with PCOS on studied parameters.

Parameters	Women with PCOS		
	WHR < 0.8 n = 54	WHR ≥ 0.8 n = 34	*p* Value
Age (years)	23.86 ± 4.83 23.00 (20.00–27.00)	25.21 ± 3.93 25.00 (22.00–27.00)	NS
BMI (kg/m^2^)	23.13 ± 4.75 21.72 (20.44–24.68)	29.43 ± 6.73 30.46 (25.95–34.29)	0.000
WHR	0.73 ± 0.04 0.74 (0.70–0.76)	0.88 ± 0.80 0.86 (0.82–0.90)	0.000
Cotinine (ng/mL)	17.30 ± 24.43 0.41 (0.02–37.86)	15.21 ± 23.86 0.39 (0.08–24.94)	NS
mF-G (0–36)	4.66 ± 4.59 4.00 (1.00–7.00)	7.00 ± 4.39 8.00 (3.00–10.00)	NS
Acne (1–5)	3.24 ± 10.71 2.00 (1.00–2.00)	4.15 ± 14.92 2.00 (1.00–2.00)	NS
Apelin (pg/mL)	36.05 ± 9.13 33.84 (30.11–37.81)	42.30 ± 55.26 31.95 (27.76–35.13)	NS (*p* = 0.059)
Chemerin (ng/L)	121.92 ± 50.62 117.82 (78.81–147.53)	137.11 ± 66.76 122.09 (89.96–178.07)	NS
Lipocalin-2 (ng/mL)	214.43 ± 88.77 206.39 (144.31–272.71)	221.65 ± 79.35 213.81 (160.30–284.85)	NS
Cholesterol (mg/dL)	158.94 ± 28.21 154.00 (137.00–178.00)	169.53 ± 32.67 165.00 (146.00–195.00)	NS
HDL-C (mg/dL)	59.62 ± 11.25 57.60 (51.30–66.50)	49.45 ± 11.48 50.40 (41.10–58.10)	0.000
LDL-C (mg/dL)	82.23 ± 24.36 79.00 (65.60–100.00)	94.00 ± 25.21 90.84 (72.80–116.20)	0.026
Triglycerides (mg/dL)	85.65 ± 37.19 77.05 (57.20–106.00)	130.44 ± 71.14 111.00 (73.80–157.00)	0.001
Glucose 0’ (mg/dL)	84.84 ± 5.11 84.35 (81.00–87.60)	85.32 ± 7.61 84.65 (79.80–89.40)	NS
Insulin 0’ (mU/mL)	6.52 ± 2.55 6.14 (4.56–7.38)	12.72 ± 8.43 11.75 (6.89–16.00)	0.000
HOMA-IR	1.37 ± 0.57 1.27 (0.93–1.58)	2.84 ± 2.10 2.41 (1.60–3.40)	0.000
LH (lU/L)	9.26 ± 7.61 7.15 (5.75–9.80)	7.96 ± 4.13 7.21 (5.27–10.10)	NS
FSH (lU/L)	6.40 ± 1.28 6.25 (5.64–7.59)	5.81 ± 1.09 5.68 (5.31–6.31)	0.011
DHEA-S (µg/mL)	301.24 ± 127.73 278.00 (222.00–323.00)	350.15 ± 159.63 314.00 (218.00–446.00)	NS
SHBG (nmol/L)	61.08 ± 26.60 57.10 (42.40–72.70)	43.48 ± 40.02 29.90 (21.60–47.30)	0.000
tTest (ng/mL)	0.36 ± 0.16 0.33 (0.25–0.43)	0.46 ± 0.20 0.44 (0.32–0.62)	0.003
fTest (pg/mL)	2.12 ± 1.50 1.71 (1.20–2.51)	4.32 ± 3.38 3.49 (1.60–5.68)	0.000
AD (ng/mL)	2.35 ± 0.81 2.18 (1.77–2.57)	2.67 ± 1.18 2.46 (1.91–3.21)	NS
FAI	2.53 ± 1.96 2.01 (1.43–2.82)	5.97 ± 4.49 4.18 (3.33–8.26)	0.000
AMH (ng/mL)	5.97 ± 3.34 5.67 (3.24–8.47)	6.87 ± 3.70 6.58 (4.36–7.73)	NS

Legend: PCOS—polycystic ovary syndrome; BMI—body mass index; mF-G—Ferriman–Gallwey scale; HDL-C—high-density lipoprotein; LDL-C—low-density lipoprotein; HOMA-IR—homeostatic model assessment for insulin resistance; LH—luteinizing hormone; FSH—follicle-stimulating hormone; DHEA—S-dehydroepiandrosterone sulfate; SHBG—sex hormone-binding globulin; tTest—total testosterone; fTest—free testosterone; AD—androstenedione; FAI—free androgen index; AMH—anti-Müllerian hormone, NS—not significant.

**Table 5 ijms-25-08713-t005:** Correlation coefficients between peptide hormones in the entire group of women with PCOS.

Correlated Parameters	Entire Group of Women with PCOS		
	Apelin [pg/mL]	Chemerin [ng/L]	Lipocalin-2 [ng/mL]
Age (years)	NS	NS	NS
BMI (kg/m^2^)		r = 0.23; *p* < 0.035	NS
WHR		r = 0.26; *p* < 0.017	NS
Cotinine (ng/mL)		NS	r = 0.29; *p* < 0.007
mF-G (0–36)		NS	NS
Acne (1–5)		NS	NS
Cholesterol (mg/dL)		NS	NS
HDL-C (mg/dL)		r = −0.24; *p* < 0.028	r = −0.32; *p* < 0.003
LDL-C (mg/dL)		NS	NS
Triglycerides (mg/dL)		r = 0.23; *p* < 0.036	r = 0.37; *p* < 0.001
Glucose 0’ (mg/dL)		r = 0.27; *p* < 0.013	NS
Insulin 0’ (mU/mL)		r = 0.24; *p* < 0.029	NS
HOMA-IR		NS	r = 0.30; *p* < 0.006
LH (lU/L)		NS	NS
FSH (lU/L)		NS	r = −0.26; *p* < 0.016
DHEA-S (µg/mL)		NS	r = 0.36; *p* < 0.001
SHBG (nmol/L)		NS	r = −0.43; *p* < 0.000
tTest (ng/mL)		NS	r = 0.32; *p* < 0.002
fTest (pg/mL)		NS	r = 0.33; *p* < 0.002
AD (ng/mL)		NS	NS
FAI		NS	r = 0.44; *p* < 0.001
AMH (ng/mL)		NS	NS

Legend: PCOS—polycystic ovary syndrome; BMI—body mass index; mF-G—Ferriman–Gallwey scale; HDL-C—high-density lipoprotein; LDL-C—low-density lipoprotein; HOMA-IR—homeostatic model assessment for insulin resistance; LH—luteinizing hormone; FSH—follicle-stimulating hormone; DHEA—S-dehydroepiandrosterone sulfate; SHBG—sex hormone-binding globulin; tTest—total testosterone; fTest—free testosterone; AD—androstenedione; FAI—free androgen index; AMH—anti-Müllerian hormone, NS—not significant.

**Table 6 ijms-25-08713-t006:** Correlation coefficients between peptide hormones in the smoking group of women with PCOS.

Correlated Parameters	Smoking Women with PCOS		
	Apelin [pg/mL]	Chemerin [ng/L]	Lipocalin-2 [ng/mL]
Age (years)	NS	NS	NS
BMI (kg/m^2^)	r = −0.45; *p* < 0.012	NS	NS
WHR	NS	r = 0.36; *p* < 0.048	NS
Cotinine (ng/mL)	NS	NS	NS
mF-G (0–36)	NS	NS	r = −0.46; *p* < 0.009
Acne (1–5)	NS	NS	NS
Cholesterol (mg/dL)	NS	r = 0.42; *p* < 0.019	NS
HDL-C (mg/dL)	NS	r = −0.44; *p* < 0.013	NS
LDL-C (mg/dL)	NS	r = 0.50; *p* < 0.004	NS
Triglycerides (mg/dL)	NS	r = 0.45; *p* < 0.010	NS
Glucose 0’ (mg/dL)	NS	r = 0.40; *p* < 0.026	NS
Insulin 0’ (mU/mL)	r = −0.38; *p* < 0.035	r = 0.38; *p* < 0.034	NS
HOMA-IR	r = −0.39; *p* < 0.029	r = 0.42; *p* < 0.019	NS
LH (lU/L)	NS	NS	NS
FSH (lU/L)	NS	NS	r = −0.51; *p* < 0.003
DHEA-S (µg/mL)	NS	NS	NS
SHBG (nmol/L)	r = 0.39; *p* < 0.028	r = −0.39; *p* < 0.031	r = −0.43; *p* < 0.017
tTest (ng/mL)	NS	NS	NS
fTest (pg/mL)	NS	r = 0.40; *p* < 0.029	NS
AD (ng/mL)	NS	NS	r = 0.40; *p* < 0.026
FAI	NS	r = 0.36; *p* < 0.045	NS
AMH (ng/mL)	NS	NS	NS

Legend: PCOS—polycystic ovary syndrome; BMI—body mass index; mF-G—Ferriman–Gallwey scale; HDL-C—high-density lipoprotein; LDL-C—low-density lipoprotein; HOMA-IR—homeostatic model assessment for insulin resistance; LH—luteinizing hormone; FSH—follicle-stimulating hormone; DHEA—S-dehydroepiandrosterone sulfate; SHBG—sex hormone-binding globulin; tTest—total testosterone; fTest—free testosterone; AD—androstenedione; FAI—free androgen index; AMH—anti-Müllerian hormone, NS—not significant.

## Data Availability

The data presented in this study are available upon request from the corresponding author.

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
