# Peer review of "Impact of Smoking and Obesity on the Selected Peptide Hormones and Metabolic Parameters in the Blood of Women with Polycystic Ovary Syndrome—Preliminary Study"

_ijms, 2024, doi:10.3390/ijms25168713_

Round 1

Reviewer 1 Report

Comments and Suggestions for Authors

Dear authors,

Firstly, congratulations on choosing the theme. Below are some notes.

1) Abstract: Review the writing. It got a little confusing. I had the impression that the result had given and as I read it, the impression changed. It is necessary to get straight to the point.

2) Introduction:

a) Reescrever o início, os textos estão meio soltos.

b) Line 55 - Chemerin can influence progesterone levels. As? Chemerina high or low?

c) I was unable to understand the objective of this study, however I understood something very similar previously in another article.

d) Rewrite the beginning, the texts are a little loose.

3) Methodology

The article came without the methodology, which makes evaluation unfeasible.

4) There are also several similar excerpts in previous studies, carried out by the same group of researchers.

Comments on the Quality of English Language

Minor English language editing required

Author Response

REVIEWER #1

Firstly, congratulations on choosing the theme. Below are some notes.

We thank the Reviewer for taking the time to read, assess the content of the manuscript and such motivating feedback.

Comments 1: Abstract: Review the writing. It got a little confusing. I had the impression that the result had given and as I read it, the impression changed. It is necessary to get straight to the point.

Response 1: Thank you very much for your suggestion. We  rewritten the entire abstract. Below please find improved version as well as in manuscript changed abstract is marked using blue colour:

“We investigated the effects of tobacco smoke exposure and abnormal body weight on selected peptide hormones and their association with metabolic and hormonal disorders in women with polycystic ovary syndrome (PCOS). The study group included 88 women with PCOS and 28 women without the disease.

In women with PCOS, chemerin, lipocalin, and apelin concentrations were influenced by overweight and obesity, with the highest concentrations observed in those with a BMI≥30.0. Exposure to tobacco smoke significantly increased only lipocalin-2 concentration. The disease itself did not affect the concentrations chemerin, lipocalin, and apelin. Additionally, we found a positive correlation between chemerin concentration and fating glucose, fasting insulin, and triglycerides levels, while a negative correlation was observed with HDL-C concentration. In the smoking subgroup, chemerin concentration was positively correlated with free testosterone concentration and the free androgen index, and negatively associated with sex hormone-binding globulin concentration.

Our findings indicate that abnormal body weight has a stronger impact than tobacco smoke exposure on metabolic and hormonal disorders in women with PCOS, highlighting the important role of weight control in such individuals. However, smoking appears to be an additional factor that intensifies hormonal disorders associated with adipose tissue.“

Comments 2: Introduction:

  1. Reescrever o início, os textos estão meio soltos.
  2. Line 55 - Chemerin can influence progesterone levels. As? Chemerina high or low?
  3. I was unable to understand the objective of this study, however I understood something very similar previously in another article.
  4. Rewrite the beginning, the texts are a little loose.

Response 2: We have, accordingly rewritten the beginning of introduction section.  Below please find a new version, all changes are marked using blue colour:

“The issue of female infertility in one of the main causes of decreased population growth in developed countries worldwide [1]. One of the leading causes of female infertility is polycystic ovary syndrome (PCOS), which affects 8–13% of women of reproductive age [2,3], and presents both fertility and metabolic disorders [4]. Epidemiological studies show that approximately 50% of women with PCOS experience obesity and metabolic syndrome [5,6]. Additionally, obesity exacerbates PCOS [7]. Women with PCOS have a higher prevalence of conditions such as insulin resistance (IR) with a prevalence of 35-80%, type 2 diabetes (T2D), cardiovascular diseases, and infertility compared to the general female population [8,9], which can also be intensified by obesity.

Adipose tissue is a highly active endocrine organ that, through the action of secreted hormone-like compounds, can influence hormonal and metabolic status [10]. Among the molecules predominantly released by adipocytes are chemerin [11], apelin [12] and lipocalin-2 [13], which are involved in many metabolic and hormonal processes.

Previous studies have shown that chemerin, apelin, and lipocalin-2 regulate insulin secretion, with changes in their levels observed in patients with diabetes [14–16]. Additionally, chemerin and lipocalin-2 participate in inflammatory processes [17,18], and the levels of chemerin and apelin may be associated with sex hormones homeostasis. An experimental study conducted on female mice showed that chemerin/G protein-coupled receptor 1 signaling regulates the secretion of prostaglandin-induced progesterone by ovarian cells during follicular development, corpus luteum formation, and luteolysis [19]. Furthermore, increased chemerin concentration can decrease insulin-like growth factor-1-stimulated progesterone and estradiol production in human granulosa cells [20], while apelin regulates the endocrine function of the human placenta [21]. Incorrect expression of lipocalin-2 is related to diseases of the reproductive system [22]. In the female reproductive system, lipocalin-2 plays a pivotal role in tissue reorganization during the menstrual cycle and pregnancy, with higher concentrations observed in the plasma of pregnant women, especially those who are overweight or obese, correlating with markers of IR [23]. As a multifaceted protein, chemerin is expressed in the ovary, placenta, and testis, indicating its potential involvement in normal female reproduction as well as in reproductive system disorders, including PCOS [24]. However, the exact mechanisms of apelin, lipocalin-2, and chemerin, as well as the specific cells that produce them in the course of PCOS, remain unidentified.”

Furhermore, in the introduction we explained the relationships between the concentration of chemerin and progesteron:

“An experimental study conducted on female mice showed that chemerin/G protein-coupled receptor 1 signaling regulates the secretion of prostaglandin-induced progesterone by ovarian cells during follicular development, corpus luteum formation, and luteolysis [19]. Furthermore, increased chemerin concentration can decrease insulin-like growth factor-1-stimulated progesterone and estradiol production in human granulosa cells [20], while apelin regulates the endocrine function of the human placenta [21].

Furthermore, we improved the goal of our study as well as the conclusions.

“The goal of the current study is to investigate the effect of exposure to tobacco smoke and abnormal body weight on selected peptide hormones and their association with metabolic and hormonal parameters in women with PCOS. To the best of our knowledge, the effect of exposure to tobacco smoke on the concentration of chemerin, apelin, and lipocalin-2 in women with PCOS has not yet been evaluated.”

“Our findings indicate that abnormal body weight has a stronger impact than tobacco smoke exposure on metabolic and hormonal disorders in women with PCOS, highlighting the important role of weight control in such individuals. However, smoking appears to be an additional factor that intensifies hormonal disorders associated with adipose tissue.

Furthermore, our results clearly confirm the complexity and heterogeneous nature of PCOS and the difficulty in unambiguously estimating the impact of individual etiological factors in the course of this disease. 

We also highlight that, due to the relatively small number of patients, our results should be treated as a preliminary.”

We also introduced the Figure 1 to summarize the main results observed in the entire group of women with PCOS.

Comments 3: Methodology. The article came without the methodology, which makes evaluation unfeasible.

Response 3: Thank you for pointing this out. We have introduced an additional information in Material & methods section. New information are marked using blue colour. Please find it below and in the manuscript on page 12.

„4. Materials and Methods

The study group will consist of 116 women, of whom 88 were diagnosed with PCOS, while 28 women were identified as not having this disease. The diagnosis of PCOS was established based on the Rotterdam criteria, requiring at least two of the following three conditions: the ovulatory dysfunction (OD) (menstrual cycles longer than 35 days or amenorrhea), clinical or biochemical signs of hyperandrogenism (HA), and polycystic ovarian morphology (PCOM) on ultrasound. The diagnosis was made after ruling out other causes of hyperandrogenism, such as congenital adrenal hyperplasia, Cushing’s syndrome, or virilization. Additionally, four different phenotypes of PCOS were recognized according to the Rotterdam criteria: phenotype 1 - with OD, HA, PCOM; phenotype 2 – with OD, HA; phenotype 3 - with HA, PCOM, and phenotype 4 - with OD, PCOM. The frequency of these phenotypes was: 65% for phenotype 1; 10% for phenotype 2; 19% for phenotype 3, and 6% for phenotype 4.

Within the PCOS patient group, women were further divided based on their exposure to tobacco smoke: 57 were non-smokers, and 31 were smokers. Biological material for the research was obtained through collaboration with the Department of Gynecological Endocrinology at the Silesian Medical University in Katowice, as part of the research project titled "Molecular Studies Useful in the Assessment of Metabolic Disorders in Patients with Polycystic Ovary Syndrome", Registration Number CWN UMW: BW-52/2020, established on May 17, 2020. Ethical approval for the research was obtained from the Bioethical Committee of Wroclaw Medical University, Poland (KBN No. 254/2021), and was conducted in accordance with the Helsinki Declaration.

The blood samples were collected during the follicular phase (within 3 and 5 days of the menstrual cycle).”

Comments 4: There are also several similar excerpts in previous studies, carried out by the same group of researchers.

Response 4: It is a new group of women with PCOS collected in 2022 years. During hospitalization in the Department of Gynecological Endocrinology at the Silesian Medical University in Katowice, standard parameters for PCOS diagnosis were determined to evaluate the relationships between disorders presented in the course of PCOS, including insulin resistance, hormonal, and metabolic disorders, and antrophometric measuremts. We must show these characteristics alongside the studied parameters.

We also highlighted in the manuscript that this is a continuation of our studies conducted on women with PCOS, as well as analyzing the effect of exposure to tobacco smoke on female fertility. Please find this information below and on the page 2, lines: 75-78 in manuscript.

“This study builds on previous research conducted on women with PCOS, which demonstrated significant associations between metabolic and hormonal disorders [29], abdominal obesity [30], adipose tissue distribution [31], oxidative stress severity [32], and the harmful effects of tobacco smoke on female fertility [33–36].”

Comments 5: English language should be checked, and editing done to ensure clarity of expression.

Response 5: English language was improved in whole text and all changes are highlighted by using blue colored text.

Reviewer 2 Report

Comments and Suggestions for Authors

Overall article is very interesting and provides valuable insights into PCOS and specific biomarkers, and well-designed & well written.

However, the diagnostic criteria in the background needs to be added, such as comparing the Rotterdam criteria versus the NIH criteria versus Androgen Excess PCOS Society criteria. Additionally, writers should mention which criteria they used for this study to diagnose PCOS. 

Small study group, in the future studies please consider larger sample size and studying other inflammatory markers such as IL-6. 

Author Response

REVIEWER #2

Comments 1: Overall article is very interesting and provides valuable insights into PCOS and specific biomarkers, and well-designed & well written.

Response 1: We thank the Reviewer for taking the time to read, assess the content of the manuscript and such motivating feedback.

Comments 2: However, the diagnostic criteria in the background needs to be added, such as comparing the Rotterdam criteria versus the NIH criteria versus Androgen Excess PCOS Society criteria. Additionally, writers should mention which criteria they used for this study to diagnose PCOS. 

Response 2: Thank you for pointing this out. We have introduced an additional information in Material & methods section. New information are marked using blue colour. Please find it below and in the manuscript on page 12.

„4. Materials and Methods

The study group will consist of 116 women, of whom 88 were diagnosed with PCOS, while 28 women were identified as not having this disease. The diagnosis of PCOS was established based on the Rotterdam criteria, requiring at least two of the following three conditions: the ovulatory dysfunction (OD) (menstrual cycles longer than 35 days or amenorrhea), clinical or biochemical signs of hyperandrogenism (HA), and polycystic ovarian morphology (PCOM) on ultrasound. The diagnosis was made after ruling out other causes of hyperandrogenism, such as congenital adrenal hyperplasia, Cushing’s syndrome, or virilization. Additionally, four different phenotypes of PCOS were recognized according to the Rotterdam criteria: phenotype 1 - with OD, HA, PCOM; phenotype 2 – with OD, HA; phenotype 3 - with HA, PCOM, and phenotype 4 - with OD, PCOM. The frequency of these phenotypes was: 65% for phenotype 1; 10% for phenotype 2; 19% for phenotype 3, and 6% for phenotype 4.

Within the PCOS patient group, women were further divided based on their exposure to tobacco smoke: 57 were non-smokers, and 31 were smokers. Biological material for the research was obtained through collaboration with the Department of Gynecological Endocrinology at the Silesian Medical University in Katowice, as part of the research project titled "Molecular Studies Useful in the Assessment of Metabolic Disorders in Patients with Polycystic Ovary Syndrome", Registration Number CWN UMW: BW-52/2020, established on May 17, 2020. Ethical approval for the research was obtained from the Bioethical Committee of Wroclaw Medical University, Poland (KBN No. 254/2021), and was conducted in accordance with the Helsinki Declaration.

The blood samples were collected during the follicular phase (within 3 and 5 days of the menstrual cycle).”

Comments 3: Small study group, in the future studies please consider larger sample size and studying other inflammatory markers such as IL-6. 

Response 3: Thank you very much. We also agree with this comment and, in the future, we will try to perform further investigations on larges groups as well as we will extend the studied parameters.

To highlighted that the study was conducted in the relatively small groups of women we added to the title information that this study should be treated as a preliminary study:

„ Impact of smoking and obesity on the selected peptide hormones and metabolic parameters in the blood of women with polycystic ovary syndrome - preliminary study”.

We also added at the end of discussion section information: “We also highlighted that, due to the relatively small number of patients, our results should be treated as a preliminary.”

We also introduced the Figure 1 to summarize the main results observed in the entire group of women with PCOS.

Reviewer 3 Report

Comments and Suggestions for Authors

The authors are attempting to decipher the link between metabolic diseases and PCOS with adipokines which is another relatively novel method to explore the risks to health to PCOS women.

1. More can be described on how the mice study and human study designs are utilized in this manuscript, importantly to how to identify these novel markers which could predict PCOS women's risks of metabolic diseases for earlier intervention

2. The analyses seemed a bit haphazard as the focus of the associations are not evident and I would like the rationale and tabled analyses to be systematically listed so that the analyses and associations with the selected biomarkers will be clearer.

3. In view of the above, the conclusion/summary should be rewritten to suit the data better. It is also envisaged that the authors will propose a potential manuscript with stronger emphasis on PCOS women's reproductive health - this is still research?

Comments on the Quality of English Language

English language should be checked, and editing done to ensure clarity of expression.

Author Response

REVIEWER #3

The authors are attempting to decipher the link between metabolic diseases and PCOS with adipokines which is another relatively novel method to explore the risks to health to PCOS women.

We thank the Reviewer for taking the time to read, assess the content of the manuscript and give valuable suggestions.

Comments 1: More can be described on how the mice study and human study designs are utilized in this manuscript, importantly to how to identify these novel markers which could predict PCOS women's risks of metabolic diseases for earlier intervention

Response 1: Thank you very much for your valuable suggestion. We have added new studies describing experimental and clinical research:

  • Wang, Q.; Kim, J.Y.; Xue, K.; Liu, J.; Leader, A.; Tsang, B.K. Chemerin, a Novel Regulator of Follicular Steroidogenesis and Its Potential Involvement in Polycystic Ovarian Syndrome. Endocrinology 2012, 153, 5600–5611, doi:10.1210/en.2012-1424.
  • Choi, Y.S.; Yang, H.I.; Cho, S.; Jung, J.A.; Jeon, Y.E.; Kim, H.Y.; Seo, S.K.; Lee, B.S. Serum Asymmetric Dimethylarginine, Apelin, and Tumor Necrosis Factor-α Levels in Non-Obese Women with Polycystic Ovary Syndrome. Steroids 2012, 77, 1352–1358, doi:10.1016/j.steroids.2012.08.005.
  • Halawa, M.R.; Ali Hendawy, L.M.; Makram, M.A. Lipocalin-2 Level in Patients with Polycystic Ovary Syndrome: Association with Insulin Resistance and Metformin Therapy. QJM: An International Journal of Medicine 2024, 117, hcae070.318, doi:10.1093/qjmed/hcae070.318.

and discussed them in relations to our results. Please find the new parts in the Discussion section in lines 229-241; 250-262, and 265-269:

 “and Wang et. al. [41]. Lima et al. [40] noted that increased serum chemerin levels were evident in overweight rats, suggesting that the concentration of chemerin detected in serum may be reflective of a systemic response and associated with obesity. Wang et all. [41], in the study conducted on a hyperandrogenic rodent model, showed that recombinant chemerin suppressed FSH-induced expression of steroidogenic enzymes, including aromatase, a crucial enzyme catalyzing the conversion of androgens into estrogens. Therefore changes in chemerin concentration could be associated with hyperandrogenism, which is one of the main feature of PCOS. Summarizing the involvement of chemerin in the course of PCOS, we can suppose that the concentration of chemerin is affected by disorders in lipid and glucose profiles and the presence of hyperandrogenism. Exposure to tobacco smoke could be recognized as an additional factor influencing these relationships, but not as effectively as the factors mentioned above.”

“Even though we detected significant differences in apelin concentration, we did not reveal any significant correlations between apelin concentration and sex hormone levels. In contrast, the clinical study performed by Choi et al. [42] on 115 non-obese (BMI < 25 kg/m2) Korean women with PCOS found that serum apelin concentration was negatively correlated with tTest levels and the FAI, and these relationships were found to be independent of IR. The differences between our studies might be associated with the characteristics of the studied groups – in Choi’s study, only women with normal body weight were investigated, while in our study approximately 47% women were with abnormal weight (BMI≥25.0 or BMI≥30.0). Additionally, our study also included smoking women. Finally, other crucial factors could be related to ethnic, environmental and geographic differences, which could also affects the investigated parameters. Therefore, further investigations performed on larges groups are needed to better explain the role of apelin in the course of PCOS.”

“Conversely, research conducted by Panidis et al. [44], Koiou et al. [45] and Halawa et al. [46] did not find any significant changes between women with and without PCOS. However, these studies [44–46] noted that weight loss significantly reduced lipcalin-2 levels in overweight or obese women with PCOS, including those taking metformin [46].”

All changes are highlighted by using blue colored text.

Comments 2: The analyses seemed a bit haphazard as the focus of the associations are not evident and I would like the rationale and tabled analyses to be systematically listed so that the analyses and associations with the selected biomarkers will be clearer.

Response 2: We tried to improve the entire manuscript and have rewritten the abstract (please see page 1, lines: 15-31 or below) and the beginning of the Introduction section (please see pages 1-2, lines: 35-68 or below).

“Abstract: We investigated the effects of tobacco smoke exposure and abnormal body weight on selected peptide hormones and their association with metabolic and hormonal disorders in women with polycystic ovary syndrome (PCOS). The study group included 88 women with PCOS and 28 women without the disease.

In women with PCOS, chemerin, lipocalin, and apelin concentrations were influenced by overweight and obesity, with the highest concentrations observed in those with a BMI≥30.0. Exposure to tobacco smoke significantly increased only lipocalin-2 concentration. The disease itself did not affect the concentrations chemerin, lipocalin, and apelin. Additionally, we found a positive correlation between chemerin concentration and fating glucose, fasting insulin, and triglycerides levels, while a negative correlation was observed with HDL-C concentration. In the smoking subgroup, chemerin concentration was positively correlated with free testosterone concentration and the free androgen index, and negatively associated with sex hormone-binding globulin concentration.

Our findings indicate that abnormal body weight has a stronger impact than tobacco smoke exposure on metabolic and hormonal disorders in women with PCOS, highlighting the important role of weight control in such individuals. However, smoking appears to be an additional factor that intensifies hormonal disorders associated with adipose tissue.”

“The issue of female infertility in one of the main causes of decreased population growth in developed countries worldwide [1]. One of the leading causes of female infertility is polycystic ovary syndrome (PCOS), which affects 8–13% of women of reproductive age [2,3], and presents both fertility and metabolic disorders [4]. Epidemiological studies show that approximately 50% of women with PCOS experience obesity and metabolic syndrome [5,6]. Additionally, obesity exacerbates PCOS [7]. Women with PCOS have a higher prevalence of conditions such as insulin resistance (IR) with a prevalence of 35-80%, type 2 diabetes (T2D), cardiovascular diseases, and infertility compared to the general female population [8,9], which can also be intensified by obesity.

Adipose tissue is a highly active endocrine organ that, through the action of secreted hormone-like compounds, can influence hormonal and metabolic status [10]. Among the molecules predominantly released by adipocytes are chemerin [11], apelin [12] and lipocalin-2 [13], which are involved in many metabolic and hormonal processes.

Previous studies have shown that chemerin, apelin, and lipocalin-2 regulate insulin secretion, with changes in their levels observed in patients with diabetes [14–16]. Additionally, chemerin and lipocalin-2 participate in inflammatory processes [17,18], and the levels of chemerin and apelin may be associated with sex hormones homeostasis. An experimental study conducted on female mice showed that chemerin/G protein-coupled receptor 1 signaling regulates the secretion of prostaglandin-induced progesterone by ovarian cells during follicular development, corpus luteum formation, and luteolysis [19]. Furthermore, increased chemerin concentration can decrease insulin-like growth factor-1-stimulated progesterone and estradiol production in human granulosa cells [20], while apelin regulates the endocrine function of the human placenta [21]. Incorrect expression of lipocalin-2 is related to diseases of the reproductive system [22]. In the female reproductive system, lipocalin-2 plays a pivotal role in tissue reorganization during the menstrual cycle and pregnancy, with higher concentrations observed in the plasma of pregnant women, especially those who are overweight or obese, correlating with markers of IR [23]. As a multifaceted protein, chemerin is expressed in the ovary, placenta, and testis, indicating its potential involvement in normal female reproduction as well as in reproductive system disorders, including PCOS [24]. However, the exact mechanisms of apelin, lipocalin-2, and chemerin, as well as the specific cells that produce them in the course of PCOS, remain unidentified.

Another crucial environment and lifestyle factor affecting human fertility is cigarette smoking.”

We also corrected the goal (please see lines 78-82 or below) and the conclusions of the study (please find lines 310-321 or below). 

“The goal of the current study is to investigate the effect of exposure to tobacco smoke and abnormal body weight on selected peptide hormones and their association with metabolic and hormonal parameters in women with PCOS. To the best of our knowledge, the effect of exposure to tobacco smoke on the concentration of chemerin, apelin, and lipocalin-2 in women with PCOS has not yet been evaluated.”

“Our findings indicate that abnormal body weight has a stronger impact than tobacco smoke exposure on metabolic and hormonal disorders in women with PCOS, highlighting the important role of weight control in such individuals. However, smoking appears to be an additional factor that intensifies hormonal disorders associated with adipose tissue.

Furthermore, our results clearly confirm the complexity and heterogeneous nature of PCOS and the difficulty in unambiguously estimating the impact of individual etiological factors in the course of this disease. 

We also highlight that, due to the relatively small number of patients, our results should be treated as a preliminary.

Figure 1 summarizes the main results observed in the entire group of women with PCOS.”

Furthermore, we improved the Material and Methods section (please find lines 334-357) and added a new Figure 1 to summarize the obtained results. 

 “4. Materials and Methods

The study group will consist of 116 women, of whom 88 were diagnosed with PCOS, while 28 women were identified as not having this disease. The diagnosis of PCOS was established based on the Rotterdam criteria [49], requiring at least two of the following three conditions: the ovulatory dysfunction (OD) (menstrual cycles longer than 35 days or amenorrhea), clinical or biochemical signs of hyperandrogenism (HA), and polycystic ovarian morphology (PCOM) on ultrasound. The diagnosis was made after ruling out other causes of hyperandrogenism, such as congenital adrenal hyperplasia, Cushing’s syndrome, or virilization. Additionally, four different phenotypes of PCOS were recognized according to the Rotterdam criteria: phenotype 1 - with OD, HA, PCOM; phenotype 2 – with OD, HA; phenotype 3 - with HA, PCOM, and phenotype 4 - with OD, PCOM. The frequency of these phenotypes was: 65% for phenotype 1; 10% for phenotype 2; 19% for phenotype 3, and 6% for phenotype 4.

Within the PCOS patient group, women were further divided based on their exposure to tobacco smoke: 57 were non-smokers, and 31 were smokers. Biological material for the research was obtained through collaboration with the Department of Gynecological Endocrinology at the Silesian Medical University in Katowice, as part of the research project titled "Molecular Studies Useful in the Assessment of Metabolic Disorders in Patients with Polycystic Ovary Syndrome", Registration Number CWN UMW: BW-52/2020, established on May 17, 2020. Ethical approval for the research was obtained from the Bioethical Committee of Wroclaw Medical University, Poland (KBN No. 254/2021), and was conducted in accordance with the Helsinki Declaration.

The blood samples were collected during the follicular phase (within 3 and 5 days of the menstrual cycle).”

All changes in the manuscript are marked using blue colour.

Comments 3: In view of the above, the conclusion/summary should be rewritten to suit the data better. It is also envisaged that the authors will propose a potential manuscript with stronger emphasis on PCOS women's reproductive health - this is still research?

Response 3: We have accordingly rewritten the summary and introduced a new Figure 1, which summarizes the main results observed in the entire group of women with PCOS.  You can find the corrected summary below and on the page 12, lines 311-321 and Figure 1 above, as a Response to Comments 2, or in the manuscript on page 12, lines 323-329. 

“Our findings indicate that abnormal body weight has a stronger impact than tobacco smoke exposure on metabolic and hormonal disorders in women with PCOS, highlighting the important role of weight control in such individuals. However, smoking appears to be an additional factor that intensifies hormonal disorders associated with adipose tissue.

Furthermore, our results clearly confirm the complexity and heterogeneous nature of PCOS and the difficulty in unambiguously estimating the impact of individual etiological factors in the course of this disease. 

We also highlight that, due to the relatively small number of patients, our results should be treated as a preliminary.

Figure 1 summarizes the main results observed in the entire group of women with PCOS.“

Comments 4: The manuscript needs to be revised for English language.

Response 4: English language was improved in whole text and all changes are highlighted by using blue colored text.
